# Evaluation of five models for constructing forest NPP-age relationships in China based on 3121 field survey samples

Peng Li[1,2], Rong Shang[1,2*], Jing M. Chen[1,3*], Mingzhu Xu[1,2], Xudong Lin[1,2], Guirui Yu[4], Nianpeng He[4], Li Xu[4]

5  [1] Key Laboratory for Humid Subtropical Eco-Geographical Processes of the Ministry of Education, School of Geographical Sciences, Fujian Normal University, Fuzhou, 350007, China
[2] Academy of Carbon Neutrality, Fujian Normal University, Fuzhou 350007, China
[3] Department of Geography and Planning, University of Toronto, Ontario, ON M5S 3G3, Canada
[4] Key Laboratory of Ecosystem Network Observation and Modeling, Institute of Geographic Sciences and Resources Research,
10  Chinese Academy of Sciences, Beijing, 100101, China

*Correspondence to*: Rong Shang (shangrong@fjnu.edu.cn) and Jing M. Chen (jing.chen@utoronto.ca)

**Abstract.** Forest net primary productivity (NPP), representing the net carbon gain from the atmosphere, varies significantly with forest age. Reliable forest NPP-age relationships are essential for forest carbon cycle modelling and prediction. These relationships can be derived from forest inventory or field survey data, but it is unclear which model is the most effective for simulating forest NPP variation with age. Here, we aim to establish NPP-age relationships for China's forests based on 3121 field survey samples. Five models, including the Semi-Empirical Mathematical (SEM) function, the Second-Degree Polynomial (SDP) function, the Logarithmic (L) function, the Michaelis-Menten (M) function, and the Γ function, were compared against field data. Results of the comparison showed that the SEM and the Γ function performed much better than the other three models. But due to the limited field survey samples at old ages, the Γ function showed a sharp decrease in NPP (decreased to almost zero) at old ages when building some forest NPP-age curves, while SEM could reasonably capture the variations of forest NPP at old ages. Considering the overall performance with currently available forest field survey samples, SEM was regarded as the optimal NPP-age model. The finalized forest NPP-age curves for five forest types in six regions of China can facilitate forest carbon cycle modelling and future projection by using the process-based model (InTEC) in China and may also be useful for other regions.

## 1 Introduction

Forests play a critical role in sequestering atmospheric carbon dioxide (Hicke et al., 2007; Liu et al., 2012; Eggleston et al., 2006; Pan et al., 2011) and mitigating climate change (Friedlingstein, 2020). Forest net primary productivity (NPP), which represents the net carbon gain from the atmosphere in the form of biomass accumulation (Fang et al., 2001; Chapin et al., 2006), constitutes a key component of the terrestrial carbon cycle (Alexandrov et al., 1999; Hasenauer et al., 2004; Zha et al., 2013; Zhao and Zhou, 2005). It varies significantly with forest age (Bond-Lamberty et al., 2004; Wang et al., 2007, 2011), generally featured by an initial increase at young ages, a maximum at a middle age, and then a gradual decline at old ages (Yu et al., 2017; He et al., 2012). The increase of forest NPP at young ages is mainly driven by fast increase of leaf area (Ryan et al., 1997; Yu et al., 2014), while the decline of NPP at old ages is primarily driven by the decrease in both gross primary productivity (GPP) and autotrophic respiration (Ra) as forests age, with GPP declining faster than Ra (Drake et al., 2011; Ryan et al., 1997, 2004; Ryan and Waring, 1992; Tang et al., 2014). These forest NPP-age variations have been integrated into process-based models such as InTEC (Integrated Terrestrial Ecosystem Carbon model) (Chen et al., 2000; Chen et al., 2003; Wang et al., 2011; Zhang et al., 2012) for modelling the forest carbon cycle, and building forest NPP-age curves as model inputs is therefore essential to facilitate forest carbon cycle modelling (Luyssaert et al., 2008; Chen et al., 2000; Zhang et al., 2012; Shang et al., 2023).

Forest NPP-age curves differ considerably for different regions and forest types due to their varied compositions and diverse growth environments (Yu et al., 2017; He et al., 2012). In Europe (Zaehle et al., 2006), Canada (Chen et al., 2003), and America (Guo et al., 1955; He et al., 2012), forest NPP-age curves have been established for different forest types or regions. However, these curves can't be directly used for China's forest carbon modelling because of regional differences in environmental

conditions. The NPP-age curves produced in previous studies have very different and sometimes inconsistent shapes, making it difficult to analyze the influence of environmental conditions on the curves of different forest types. To address these issues, some studies have tried to build forest NPP-age curves in China. Yu et al. (2017) established forest NPP-age curves for twelve major forest types in Heilongjiang province using forest inventory data and yield tables. Wang et al. (2018) derived forest NPP-age curves for nine pure forest types with different site indices within Heilongjiang province using the yield tables, biomass equations, and forest inventory data. Zheng et al. (2019) built two forest NPP-age curves separately for coniferous and broad-leaved forests in Zhejiang province using forest inventory data. But these curves are limited to the provincial level (currently only available in Heilongjiang and Zhejiang provinces), and cannot represent the diverse growth status of China's forests. Wang et al. (2011) constructed five forest NPP-age curves for five representative forest ecosystems in China, but the NPP data used to build these curves were obtained from the simulations of the BEPS (Boreal Ecosystem Productivity Simulator) model (Chen et al., 2012; Ju et al., 2006; Liu et al., 2002, 1999), not forest inventory data or field survey data. Furthermore, these curves didn't consider the significant differences in forest and climate conditions between the north and south of China and were insufficient to differentiate the north-south variations in China (Dai et al., 2011). Therefore, it is essential to develop forest NPP-age curves for the entire China with consideration of the differences in regions and forest types.

There were some models that could be used to simulate the forest NPP-age curves (Chen et al., 2003; Yu et al., 2017; He et al., 2012; Peper et al., 2001; Semenzato et al., 2011; Dalgleish et al., 2015; Tang et al., 2014). The Semi-Empirical Mathematical (SEM) function was first developed for simulating NPP-age relationships in Canada (Chen et al., 2003), America (He et al., 2012), and China (Wang et al., 2011; Yu et al., 2017; Wang et al., 2018; Zheng et al., 2019). The Second-Degree Polynomial (SDP) function, Logarithmic (L) function, Michaelis-Menten (M) function, and $\Gamma$ function were used to build the NPP-age relationships for the boreal and temporal forests (Tang et al., 2014). The L function was mainly used to construct the relationship between diameter at breast height (DBH), forest height, and forest age (Peper et al., 2001; Semenzato et al., 2011; Dalgleish et al., 2015), and it was also used to model NPP-age relationships (Tang et al., 2014) as forest NPP is related with DBH and forest height. The M function is a common mathematical model used to describe enzyme reaction kinetics (Do et al., 2022), and was also found to be suitable for relating carbon fluxes to forest age (Tang et al., 2014). The $\Gamma$ function was demonstrated to have better performance than the SDP function, L function, and M function in building the NPP-age relationships for the boreal and temporal forests (Tang et al., 2014). Different models could show diverse performance in tracking forest NPP-age curves for different forest types and regions. To facilitate the forest carbon modelling, it is crucial to compare these models in building forest NPP-age curves across diverse forest types and regions in China.

There are two objectives of this study: (1) to build forest NPP-age relationships for the entire China considering differences in regions and forest types based on forest field survey data and remote sensing data, and (2) to compare five models and determine the optimal model in building forest NPP-age relationships across China. The built forest NPP-age curves from the optimal model for different forest types and regions in China would be served as inputs of a process-based model to facilitate China's forest carbon cycle modelling and future projection.

## 2 Study Area and Data

### 2.1 Study area

China is selected as the study area, and its forests consist of five mean functional types: Evergreen Broad-leaved Forests (EBF), Evergreen Needle-leaved Forests (ENF), Deciduous Broad-leaved Forests (DBF), Deciduous Needle-leaved Forests (DNF), and Mixed Forests (MF). The five forest types were separated in building forest NPP-age curves with consideration of their different physiological and ecological characteristics (Wang et al., 2011). Except for forest cover types, climatic differences in different regions of China can also affect the forest NPP-age relationships (Li and Zhou, 2015; Song et al., 2018), so regions also need to be divided when building the forest NPP-age curves. According to China's geographical division (Fang et al., 2001), the study area was divided into six regions (Fig. 1): Northeast China (NE), North China (N), Northwest China (NW), East China (E), Southwest China (SW), and South China (S). Significant differences in forest types can be observed among different regions. Region NE (including Heilongjiang, Jilin, and Liaoning provinces) is a typical boreal forest in the world and the most significant natural forest area in China. Region N (including Beijing and Tianjin cities, and Hebei, Shanxi, and Inner Mongolia provinces) accounts for 14% of China's total forest area and is mainly composed of DBF and ENF. Regions NW (including Gansu, Ningxia, Qinghai, Shanxi, and Xinjiang provinces) only account for 2.57% of the total forest area in China. Region E (including Shanghai City and Jiangsu, Zhejiang, Anhui, Fujian, Jiangxi, Shandong, and Taiwan provinces) accounts for 14% of China's total forest area, and its forests show significant zonal characteristics. Region SW (including Yunnan, Sichuan, Xizang, Guizhou, and Chongqing provinces) is the second-largest natural forest area in China, accounting for 26% of China's total forest area and 43% of China's forest stock (Liu et al., 2021). Region S (including Henan, Hubei, Hunan, Guangdong, Guangxi, and Hainan provinces) accounts for 20% of the total forest area in China with a large proportion of planted forests.

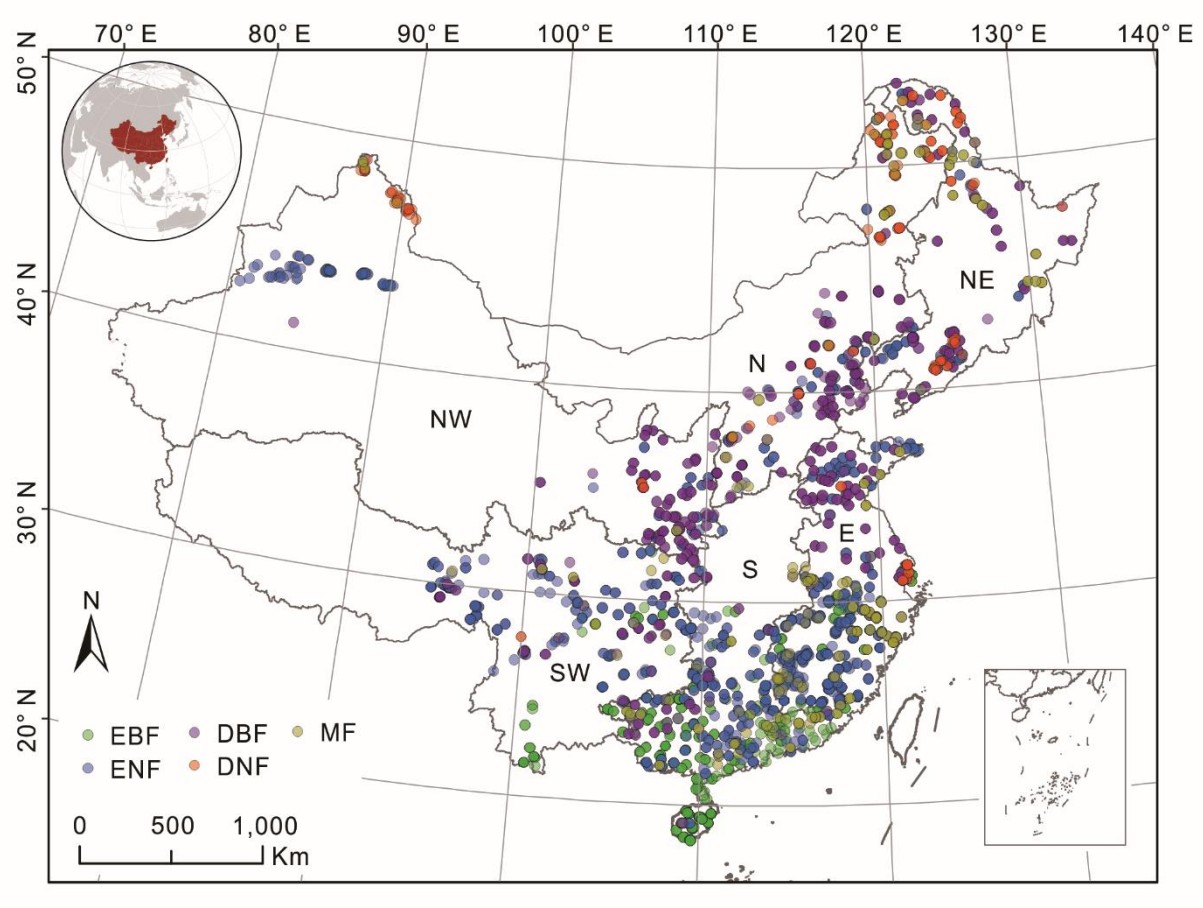

**Figure 1.** Distribution of forest field survey sites and their forest cover types (different color indicates different types) within the six regions of China. E: Northeast China; N: North China; NW: Northwest China; E: East China; S: South China; SW: Southwest China; EBF: Evergreen Broad-leaved Forests; ENF: Evergreen Needle-leaved Forests; DBF: Deciduous Broad-leaved Forests; DNF: Deciduous Needle-leaved Forests; MF: Mixed Forests.

## 2.2 Data

The forest field survey data (Fang et al., 2018) and the GLOBMAP Version 3 LAI product (Liu et al., 2012) were used to build forest NPP-age curves for different regions and forest types.

The forest field survey data includes 3121 sampling sites across China (Figure 1) except for Taiwan, Hongkong, and Macao (Fang et al., 2018). It includes 585 EBF sites, 1340 ENF sites, 745 DBF sites, 196 DNF sites, and 255 MF sites. These sites were selected according to their representativeness of the forest types in a given area, and they were sampled using the method outlined by the Intergovernmental Panel on Climate Change (IPCC) (Tang et al., 2018). This dataset records the site location, survey time (from 2009 to 2013), forest cover type, stand age, forest aboveground biomass, forest underground biomass, and so on. These attributes were first used to calculate the forest field NPP and then build the forest NPP-age curves.

The GLOBMAP Version 3 LAI product (Liu et al., 2012) was mainly used in the calculation of forest foliage biomass as part of NPP. It provides consistent long-term global leaf area index (LAI) data at 500 m spatial resolution from 1981 to 2022

on a geographical grid by fusion of Moderate Resolution Imaging Spectroradiometer (MODIS) and Advanced Very High-Resolution Radiometer (AVHRR) data. According to the site location and survey time, the annual maximum LAI within the survey year for the field survey sample was used to calculate the turnovers of foliage and turnovers of fine roots in the soil.

## 3 Methods

### 3.1 Calculating forest field NPP

Forest field NPP was not directly provided by the forest field survey data. As 33%−50% of forest NPP is allocated to foliage and fine roots each year (Gower et al., 1997), the forest field NPP was calculated from forest field survey data considering four components (Chen et al., 2002; He et al., 2012): total biomass increase (sum of the stem, branch, and coarse root biomass), mortality, turnovers of foliage, and turnovers of fine roots in the soil (Chen et al., 2002; He et al., 2012; Xia et al., 2019).

$$NPP = \triangle B_c + M + L_l + L_{fr}, \tag{1}$$

where $\triangle B_c$ is the annual increment of total living biomass including stems, branches, and coarse roots; $M$ is the mortality per year that includes standing dead trees and fallen dead trees; $L_l$ is the turnover of leaves per year; and $L_{fr}$ is the turnover of fine roots per year in the soil. Mortality (M) is ignored in this study due to a lack of observations at the ground plots and its average small proportion of NPP except for extreme conditions (See section 5.2 for detailed discussion).

The annual increment of total living biomass was calculated from the annual biomass change ($\triangle B$) and the ratio from

biomass to carbon ($c$) (White et al., 2000; Xia et al., 2019). The $c$ was set to 0.5 following previous studies (Van Tuyl et al., 2005; Fang et al., 2001; Pan et al., 2011).

$$\triangle B_c = \triangle B \times c, \tag{2}$$

The calculation of the leaf renewal rate ($L_l$) is related to leaf area index ($LAI$), specific leaf area ($SLA$), leaf turnover rate ($t_l$), and carbon content ($c$):

$$L_l = \frac{LAI}{SLA} \times t_l \times c, \tag{3}$$

The amount of fine root regeneration is closely related to the amount of leaf regeneration, and hence the proportions of NPP allocated to fine root and leaf are related:

$$L_{fr} = R_{fr,l} \times L_l, \tag{4}$$

where, $R_{fr,l}$ represents the ratio of carbon allocated to new fine roots to carbon in new leaves. Table 1 provides detailed values

for the coefficients of $SLA$, $t_l$, and $R_{fr,l}$ for different forest types (White et al., 2000). The coefficients of MF were calculated

as the average value of the other four forest cover types.

**Table 1.** The input parameters in the calculation of NPP for different forest types. SLA is the specific leaf area; $t_l$ is the foliage turnover ratio; $R_{fr,l}$ is the ratio of NPP to fine roots and leaves. EBF: Evergreen Broad-leaved Forests; ENF: Evergreen Needle-leaved Forests; DBF: Deciduous Broad-leaved Forests; DNF: Deciduous Needle-leaved Forests; MF: Mixed Forests.

| Forest Type | $SLA$ (m$^2$ kg C$^{-1}$) | $t_l$ (year$^{-1}$) | $R_{fr,l}$ (kg C kg C$^{-1}$) |
|---|---|---|---|
| EBF | 32.000 | 0.860 | 1.200 |
| ENF | 8.200 | 0.260 | 1.400 |
| DBF | 32.000 | 1.000 | 1.200 |
| DNF | 22.000 | 1.000 | 1.200 |
| MF | 23.550 | 0.780 | 1.300 |

## 145  3.2 Building forest NPP-age relationships

Five models, including the SEM function, SDP function, L function, M function, and $\Gamma$ function, were used to build the NPP-age relationships among the five forest cover types and six regions in China.

The SEM function (Chen et al., 2003; He et al., 2012) is as follows:

$$NPP(i) = a\left[1 + (b(i/c)^d - 1)/e^{(i/c)}\right], \tag{5}$$

where $a$, $b$, $c$, and $d$ are empirical coefficients to be determined from data, and $NPP(i)$ is NPP at the age of $i$.

The SDP function (Tang et al., 2014) is as follows:

$$NPP(i) = a \times i^2 + b \times i + c, \tag{6}$$

where $a$, $b$, and $c$ are empirical coefficients.

The L function (Peper et al., 2001; Semenzato et al., 2011; Dalgleish et al., 2015) is as follows:

$$NPP(i) = a[\log(i + 1)]^b, \tag{7}$$

where $a$ and $b$ are empirical coefficients.

The M function (Tang et al., 2014; Do et al., 2022) is as follows:

$$NPP(i) = a \times i/(b + i), \tag{8}$$

where $a$ and $b$ are empirical coefficients.

The $\Gamma$ function (Tang et al., 2014) is as follows:

$$NPP(i) = k_0 i^{k_1} e^{k_2 \cdot i}, \tag{9}$$

where $k_0$, $k_1$, and $k_2$ are empirical coefficients.

To reduce the influence of noises or outliers in building forest NPP-age curves, the forest field NPPs were averaged within different age bins (e.g. 3, 5, 10, or 20 years). The age bins were divided according to the number of samples in each age bin, and if there were not enough samples for some ages, larger age bins would be used. The $R^2$ and RMSE were used to determine the optimal model for building forest NPP-age curves in China, and the model with the highest $R^2$ and smallest RMSE would be regarded as optimal.

### 3.3 Determination of ten forest NPP-age curves

Fig. 2 shows the statistics of forest field survey samples according to the three age groups in China. The age group of 0−50 years had the most samples in all forest cover types and regions. The regions NE and N mainly contained DBF (highest number), ENF, DNF, and MF sites. The region NW was dominated by the samples of DBF (highest number) and ENF. The region SW has the most samples of ENF and identical samples of EBF, DBF, and MF. Region S mainly had the samples of ENF (highest number), EBF, and DBF. The samples of EBF and ENF were dominant in region E. The age group of 51−100 years had fewer samples than the group of 0−50 years. EBF samples were mainly located in Region E and Region SW. The samples of ENF were identical for all six regions. The samples of DNF, DBF, and MF were dominant in the north (NE/N/NW) regions, and a few samples of DBF and MF were located in the south (SW/S/E) regions. The age group of >100 years had the lowest number of samples. The sample of ENF was dominant in the regions of NW and S. The sample of EBF was dominant in the regions E and S, and the sample of DNF was dominant in regions NW.

In consideration of the survey sample and stand age distribution patterns, ten forest NPP-age curves were derived across the entire China. The samples of EBF were sufficient to separate three forest NPP-age curves for the north (NE/N/NW) regions, the SW region, and the S/E regions. The samples of ENF, BDF, and MF were sufficient to build two separate forest NPP-age curves for the north (NE/N/NW) and south (SW/S/E) regions. The samples of DNF were rare and mainly located in the north (NE/N/NW) regions, and there was only one forest growth curve for DNF in the entire China.

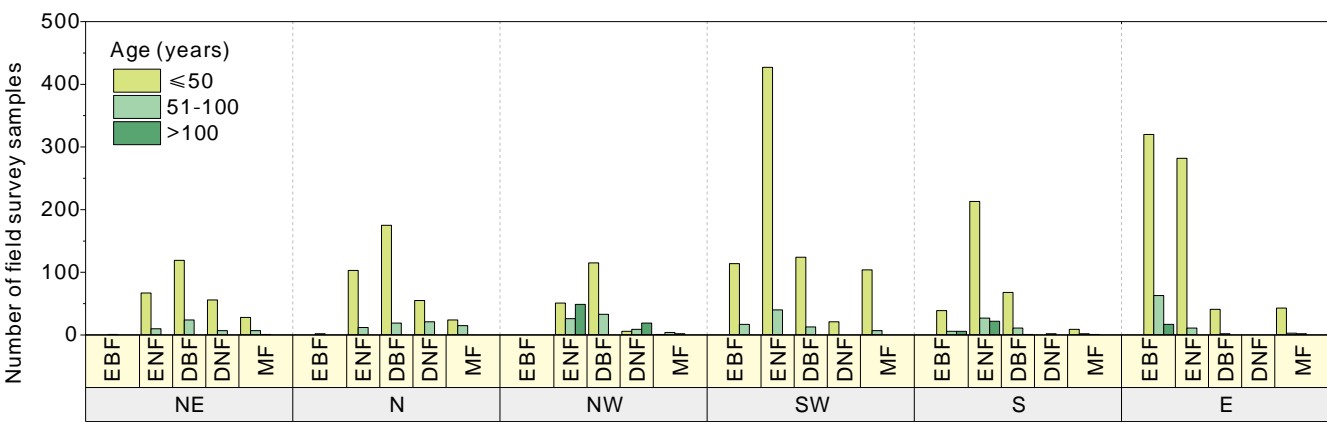

**Figure 2.** The statistics of forest field survey samples according to age groups, regions, and forest cover types in China. The first horizontal coordinate (yellow) indicates the region; the second horizontal coordinate (blue) indicates the forest cover type; the bar indicates the number

of samples; and the bar color indicates the age group (purple is for 0 to 50 years, green is for 51 to 100 years, and orange is for >100 years). E: Northeast China; N: North China; NW: Northwest China; E: East China; S: South China; SW: Southwest China; EBF: Evergreen Broad-leaved Forests; ENF: Evergreen Needle-leaved Forests; DBF: Deciduous Broad-leaved Forests; DNF: Deciduous Needle-leaved Forests; MF: Mixed Forests.

## 3.4 Uncertainty analysis

The uncertainty of an NPP-age curve mainly comes from the calculation of forest field NPP, whose uncertainty was calculated from its four components (Yu et al., 2017) in Equation (1). It was represented as the sum of the variances of four independently calculated values based on forest age group:

$$\sigma_{NPP}^2 = \sigma_{\triangle B_c}^2 + \sigma_M^2 + \sigma_{L_l}^2 + \sigma_{L_{fr}}^2, \tag{10}$$

where $\sigma_{\text{NPP}}^2$ is the uncertainty of the NPP-age curve, $\sigma_{\triangle B_c}^2$ is the uncertainty in the biomass measurements, $\sigma_M^2$ is the uncertainty in the mortality estimation, $\sigma_{L_l}^2$ and $\sigma_{L_{fr}}^2$ are the uncertainties in the estimates of the turnovers of leaves and fine roots, respectively. As $L_l$ and $L_{fr}$ were correlated, their errors were estimated as follows:

$$\sigma_{L_l + L_{fr}}^2 = \sigma_{L_l}^2 + \sigma_{L_{fr}}^2 + 2cov_{L_l, L_{fr}}, \tag{11}$$

where $\sigma_{L_l}^2$ is the standard deviation of the leaf renewal rate, $\sigma_{L_{fr}}^2$ is the standard deviation of the fine roots renewal rate, and $cov_{L_l, L_{fr}}$ is the covariance between $L_l$ and $L_{fr}$, which was simplified as $cov_{L_l, L_{fr}} \approx cov_{L_l, R_{fr,l} \cdot L_l} = R_{fr,l} \times cov_{L_l, L_l} = R_{fr,l} \times \sigma_{L_l}^2$ (He et al., 2012).

## 4 Results

## 4.1 Characterization of forest NPP-age curves

Figure 3 shows the comparison of the five models in building the NPP-age curves for various forest types and regions in China based on the averaged forest field NPP, and the three components of forest field NPP for each curve are shown in Fig. 4. The annual increment of total living biomass constitutes the predominant share of NPP, markedly surpassing the sum of other components in NPP. Despite their relatively minor proportions, the turnover rates of foliage and the fine roots in the soil are essential components of NPP (He et al., 2012). Across various forest types, the annual increment of total living biomass rises in early forest development, peaks mid-term, and later declines, generally consistent with the trajectory of NPP with age. There are also exceptions for some curves with slightly increasing trends of some NPP components in old ages. This might be explained by the following reasons: first, this study did not separate the overstory and understory LAI, and the presence and growth of understory LAI can influence the trends of the NPP components at old ages; second, due to the limited forest field survey samples, we merged samples over large regions to build the forest growth curves for some forest cover types in China,

and this can also be a reason for not showing a declining trend. Lastly, in mixed forests, the growth of different forests is asynchronous, leading to the absence of a declining trend in old ages.

**4.2 Comparison of five models in building forest NPP-age curves**

Fig. 5 shows the quantitative comparison of the five functions in building forest NPP-age curves across varied forest cover types and regions in China. The SEM function and $\Gamma$ function performed prominently in all ten curves, perfectly capturing the
NPP variations with forest age. The SEM function had the highest $R^2$ and lowest RMSE for three curves of EBF (NE/N/NW), EBF (S/E), ENF (CHN), and had the lowest RMSE but comparable $R^2$ for five curves including ENF (NE/N/NW), ENF (SW/S/E), DBF (NE/N/NW), MF (NE/N/NW), and MF (SW/S/E). The $\Gamma$ function had the highest $R^2$ and lowest RMSE for two curves of EBF (SW) and DBF (SW/S/E). The NPP-age variations were not well captured by the SDP function, L function, and M function: the declining trend of forest NPP in old ages was not captured by the L and M functions, and five constructed
curves by the SDP function exhibited unreasonable declines in NPP for older forest ages (with NPP sharply decreasing to 0 before reaching 200 years). Even though the SDP function achieved a relatively high $R^2$ (<0.05 lower than the highest $R^2$) in building two curves of EBF (SW) and DBF (SW/S/E), it had 13%−88% larger RMSE than the lowest RMSE. The M function also reached a relatively high $R^2$ (<0.05 lower than the highest $R^2$) in building four curves of EBF (S/E), ENF (NE/N/NW), DBF (NE/N/NW), and MF (NE/N/NW), but it had 29%−124% larger RMSE than the lowest RMSE.

To further evaluate the performances of the SEM function and the $\Gamma$ function, we extended the forest age to 300 years and normalized the built NPP-age curves by dividing each curve with its maximum NPP value (Fig. 6). The most significant differences between the normalized NPP-age curves simulated using these two functions appear in the extended old ages. The curves built from the SEM function exhibit stable forest NPP during old ages, while those from the $\Gamma$ function display a distinct and continuous decrease in NPP as the forests become very old. For the two curves of EBF(SW) and DBF(SW/S/E) where the
$\Gamma$ function had the highest $R^2$ and lowest RMSE, the forest NPP decreased to almost zero when the stand age reached 300 years. The forest NPP in the curves of ENF(SW/S/E), MF(SW/S/E), DBF(NE/N/NW), and DNF(CHN) built by the $\Gamma$ function also decreased sharply at the age of 300 years and decreased to almost zero in the ages of 400-500 years. These forest growth patterns contradict the results of previous studies, which indicated that forest NPP is usually reduced to about half (Mund et al., 2002; Ryan et al., 2004) or one-third (Luyssaert et al., 2008; Wang et al., 2011) of its maximum value. The curves from
the $\Gamma$ function suggest that forests would stop growth completely at old ages and act as carbon sources. However, studies have demonstrated that old forests still act as carbon sinks, despite the controversial magnitude of the forest carbon sink ranging from 1.0 to 3.2 Mg C ha$^{-1}$ yr$^{-1}$ (Gundersen et al., 2021; Luyssaert et al., 2008). Ecologically, we would expect old forests to maintain stable conditions through self-renewal processes, such as the generation of new trees after the mortality of old trees (Harmon et al., 1990). The SEM function that produces stable NPP at old ages is therefore more reasonable in capturing the
forest NPP-age variations during old ages, and was determined as the optimal forest model for building the forest NPP-age curves in China (the model coefficients of the built ten curves are provided in Table 2).

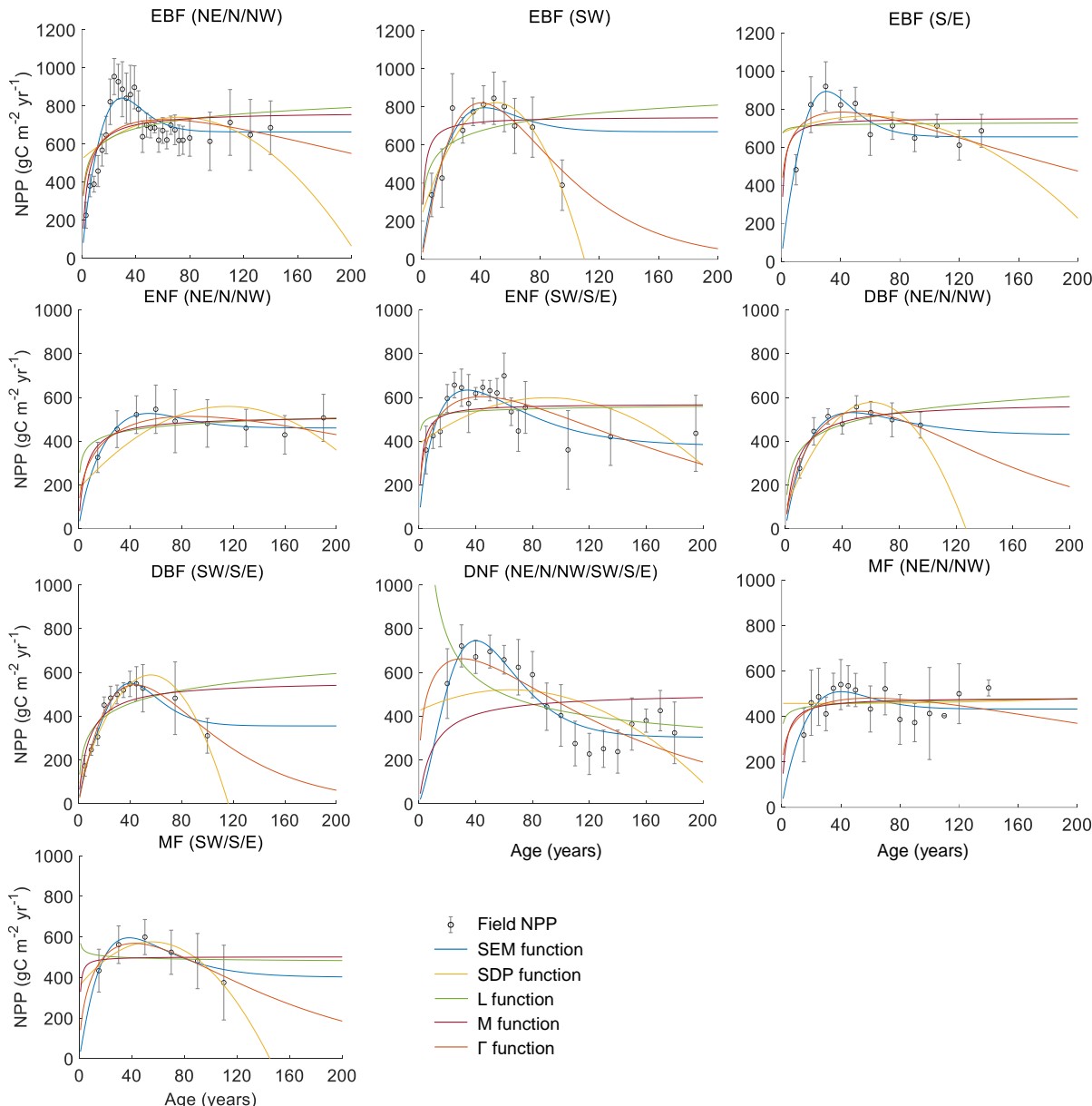

**Figure 3.** Comparing five models in building the forest NPP-age curves for different forest cover types and regions in China. In each panel, regions are shown in the upper right. The black hollow dots with error bars represent the average NPP and its one standard deviation. The five colourful lines indicate the curve-fitting from the five functions. E: Northeast China; N: North China; NW: Northwest China; E: East China; S: South China; SW: Southwest China; EBF: Evergreen Broad-leaved Forests; ENF: Evergreen Needle-leaved Forests; DBF: Deciduous Broad-leaved Forests; DNF: Deciduous Needle-leaved Forests; MF: Mixed Forests.

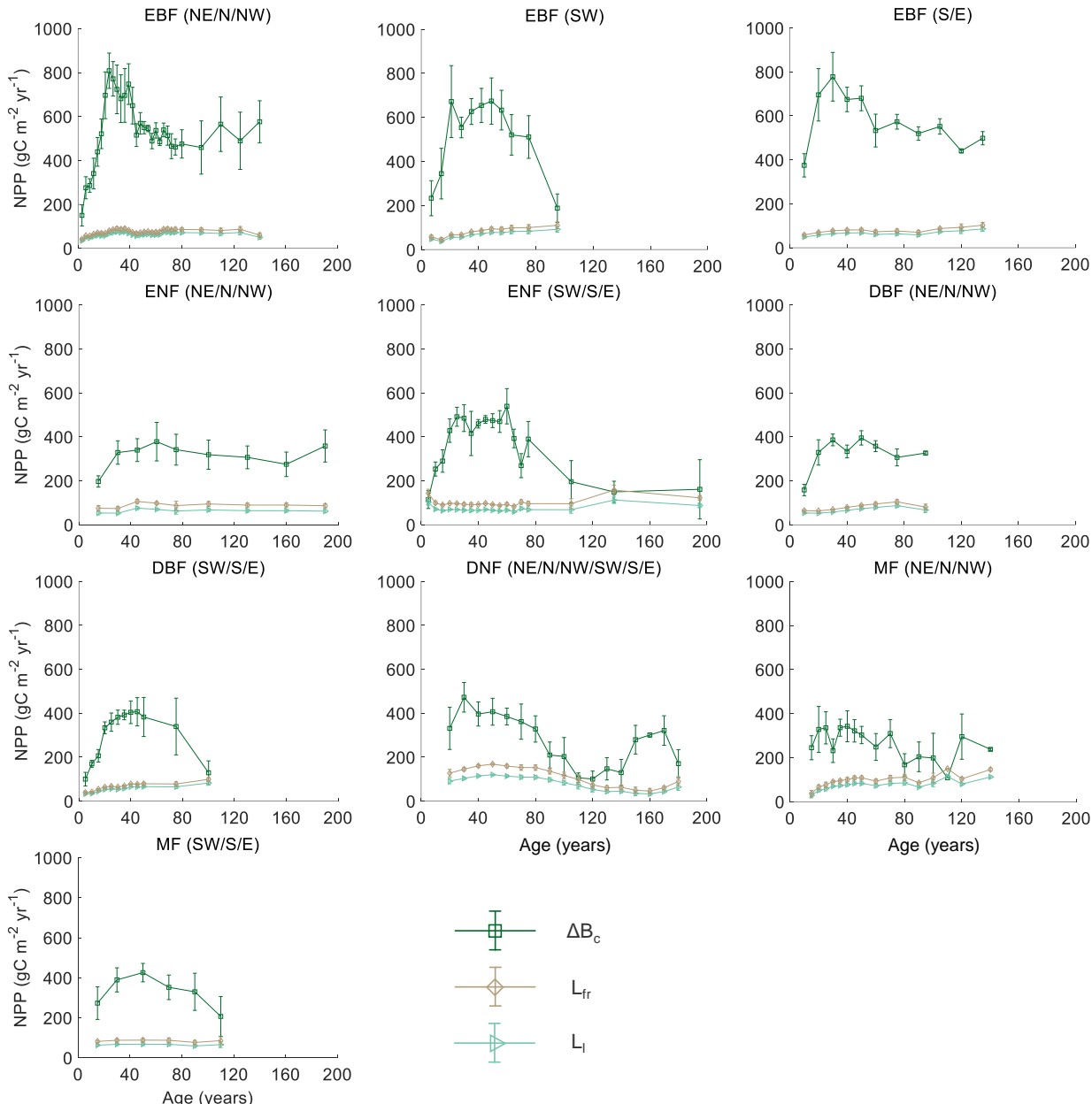

**Figure 4.** Distribution of the three components of NPP for different forest cover types and regions in China. Hollow points with error bars represent the components of NPP along with one standard deviation. $\triangle B_c$ is the annual increment of total living biomass including stems, branches, and coarse roots; $L_l$ is the turnover of leaves per year; and $L_{fr}$ is the turnover of fine roots per year in the soil. E: Northeast China; N: North China; NW: Northwest China; E: East China; S: South China; SW: Southwest China; EBF: Evergreen Broad-leaved Forests; ENF: Evergreen Needle-leaved Forests; DBF: Deciduous Broad-leaved Forests; DNF: Deciduous Needle-leaved Forests; MF: Mixed Forests.



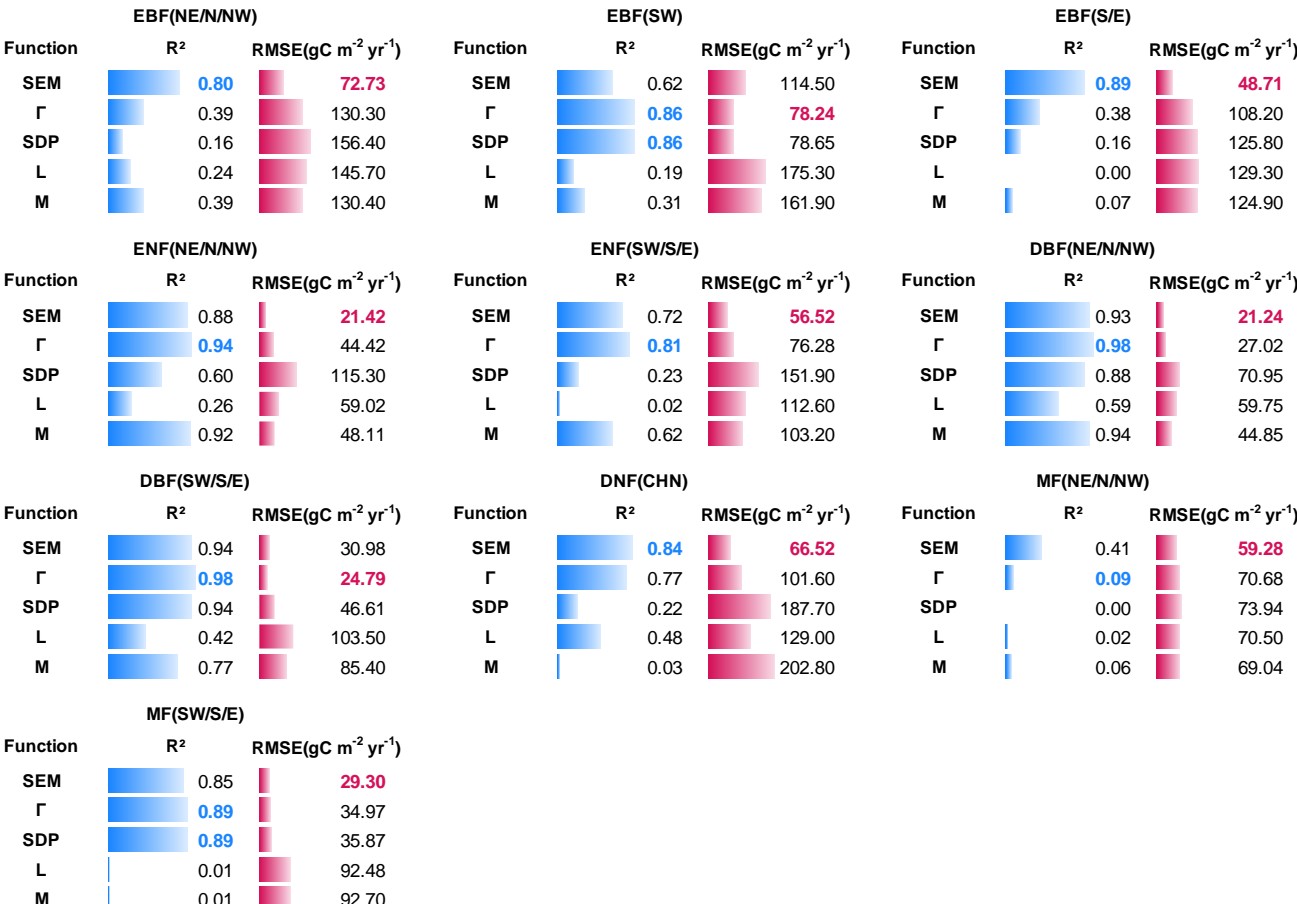

**Figure 5.** Quantitative descriptions of the five models in building the forest NPP-age curves for different forest cover types and regions in China. The highest $R^2$ is labelled with green color, and the lowest RMSE is labelled with red color. E: Northeast China; N: North China; NW: Northwest China; E: East China; S: South China; SW: Southwest China; EBF: Evergreen Broad-leaved Forests; ENF: Evergreen Needle-leaved Forests; DBF: Deciduous Broad-leaved Forests; DNF: Deciduous Needle-leaved Forests; MF: Mixed Forests.

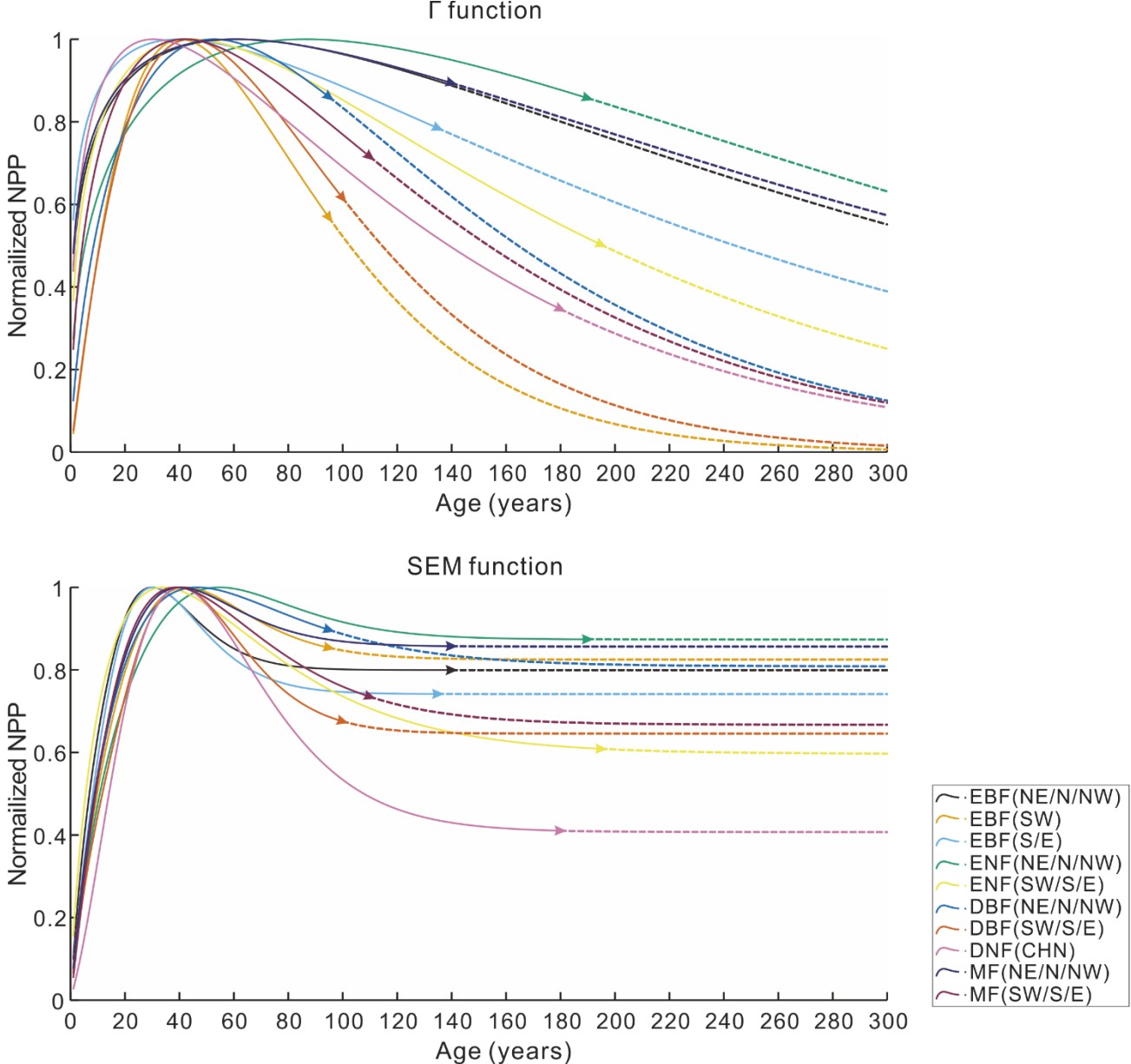

**Figure 6.** The normalized NPP-age curves built from the SEM function and the Γ function with the forest age extended to 300 years. The solid lines are for the age period with field data (the triangle in each line indicates the largest age with the field data), and the dashed lines are for the age period without field data. E: Northeast China; N: North China; NW: Northwest China; E: East China; S: South China; SW: Southwest China; EBF: Evergreen Broad-leaved Forests; ENF: Evergreen Needle-leaved Forests; DBF: Deciduous Broad-leaved Forests; DNF: Deciduous Needle-leaved Forests; MF: Mixed Forests.

**Table 2.** The coefficients of the built forest NPP-age curves by the SEM function in China (The unit of NPP calculated from the coefficients **a-d** is gC m$^{-2}$ yr$^{-1}$). **a-d**: the model coefficients; E: Northeast China; N: North China; NW: Northwest China; E: East China; S: South China; SW: Southwest China; EBF: Evergreen Broad-leaved Forests; ENF: Evergreen Needle-leaved Forests; DBF: Deciduous Broad-leaved Forests; DNF: Deciduous Needle-leaved Forests; MF: Mixed Forests.

| Forest Type | *a* | b | *c* | d |
|---|---|---|---|---|
| EBF(NE/N/NW) | 664.100 | 0.100 | 7.618 | 3.643 |
| EBF(SW) | 670.000 | 0.113 | 11.470 | 3.310 |
| EBF(S/E) | 654.600 | 0.291 | 9.131 | 3.032 |
| ENF(NE/N/NW) | 460.192 | 0.045 | 13.007 | 3.809 |
| ENF(SW/S/E) | 378.106 | 2.828 | 31.679 | 0.721 |
| DBF(NE/N/NW) | 429.184 | 1.353 | 26.832 | 0.995 |
| DBF(SW/S/E) | 355.632 | 0.097 | 9.508 | 4.153 |
| DNF(CHN) | 303.573 | 2.029 | 15.739 | 2.425 |
| MF(NE/N/NW) | 432.260 | 0.110 | 10.750 | 3.291 |
| MF(SW/S/E) | 401.900 | 1.926 | 22.778 | 1.200 |

## 4.3 Comparison to the forest NPP-age curves built previously in China

The forest NPP-age curve could be depicted by a key characteristic: the age at which forest NPP peaks (shortened as peak NPP
age). The ten built forest NPP-age curves by the SEM function in this study were compared to the forest NPP-age curves built
previously in China using this characteristic (Table 3). Climate factors have a significant influence on the peak NPP age (Zhang
et al., 2017). The NPP-age curves of forests in southern regions, characterized by higher temperatures, generally exhibit an
earlier age of peak NPP compared to forests in the northern regions with lower temperatures.

EBF achieves its highest NPP at 30 years in the regions of NE/N/NW/S/E, while this peak occurs at 42 years in region SW,
similar to the previously reported average peak age of 40 years for EBF in China (Wang et al., 2011). The peak NPP for ENF
is achieved at 55 years in the northern regions, while it occurs at 34 years in the southern regions, aligned with previous reports
where the peak age in the northern regions is 21 years later compared to the southern regions (Xu et al., 2010). But in the
southern regions, our peak NPP age of 34 years was significantly different from the 13 years reported by Wang et al. (2011).
However, their fitting points showed a significant bimodal distribution around 13 and 53 years. Considering this bimodal
distribution, the average peak NPP age could be 33 years, more closely to our findings.

DBF, predominantly located in the northern regions, peaks in NPP at the age of 47, slightly later than the southern regions
where the peak is observed at 41 years. These two values were much smaller than the 122 years reported by Wang et al. (2011),
where the NPP-age curve for DBF was built by the SDP function instead of the SEM function. This large difference for DBF
was also noticed by He et al. (2012), and our results were consistent with the peak NPP age of 27 ± 16.5 for BDFs in America
(He et al., 2012).

DNF, 60.2% located in the northern regions, reaches peak NPP at the age of 40 years, congruent with the peak growth age
derived from the same region by other researchers using the Logistic stand growth model with National Forest Inventory (NFI)
data (Xu et al., 2010). Our peak age differed by 14 years from the 54 years reported by Wang et al. (2011). However, their
fitting points demonstrated that the peak NPP spanned ages from 20 to 70 years (Wang et al., 2011), with an average of 45
years, which aligns more closely with our peak NPP age.

MF reached the peak NPP at the age of 40 in the northern regions and 39 years in the southern regions, presenting a deviation
of less than 8 years compared to Wang et al. (2011) and a consistent peak NPP reported in Heilongjiang province by Yu et al.

(2017). The peak NPP age of our national NPP-age curve shows substantial differences from the peak NPP ages identified by Yu et al. (2017) and Zheng et al. (2019) in their respective studies on the Heilongjiang and Zhejiang provinces. This could be
attributed to significant variations in forest growth patterns nationwide compared to these specific provinces, arising from various factors including but not limited to forest species, climatic conditions, and soil types (Dai et al., 2011; Zhao and Zhou, 2006; Ji et al., 2020; Xiaoyun et al., 2018).

**Table 3.** Comparison of the forest NPP-age curves built previously in China at the peak NPP age. E: Northeast China; N: North China; NW:
Northwest China; E: East China; S: South China; SW: Southwest China; EBF: Evergreen Broad-leaved Forests; ENF: Evergreen Needle-leaved Forests; DBF: Deciduous Broad-leaved Forests; DNF: Deciduous Needle-leaved Forests; MF: Mixed Forests; ENF-S: ENF in the tropics and subtropics; MBF: Mixed Broad-leaved Forests; ENF-H: ENF in Heilongjiang province, including Pinus sylvestris and Pinus koraiensis; DBF-H: DBF in Heilongjiang province, including Quercus mongolica, Planted populus, Populus davidiana, Betula davuria, Tilia, and Betula platyphylla; DNF-H: Larix gmelinii in Heilongjiang province; MBF-H: Mixed Broad-leaved Forests in Heilongjiang province;
MNF-H: Mixed Needle-leaved Forest in Heilongjiang province; MF-H: Mixed Forests in Heilongjiang province; NF-Z: Needle-leaved Forest in Zhejiang province; BF-Z: Broad-leaved Forest in Zhejiang province.

| Study area | Forest type | China regions | Methods | Age at peak NPP (year) | Source |
|---|---|---|---|---|---|
| China | EBF | NE/N/NW | SEM | 30 | Our NPP-age curves |
| | | SW | | 43 | |
| | | S/E | | 30 | |
| | ENF | NE/N/NW | | 55 | |
| | | SW/S/E | | 34 | |
| | DBF | NE/N/NW | | 47 | |
| | | SW/S/E | | 41 | |
| | DNF | CHN | | 40 | |
| | MF | NE/N/NW | | 40 | |
| | | SW/S/E | | 38 | |
| China | EBF | CHN | SEM | 40 | Wang et al. (2011) |
| | ENF-S | SW/S/E | SEM | 13 | |
| | DBF | CHN | SDP | 122 | |
| | DNF | CHN | SEM | 54 | |
| | MBF | CHN | SEM | 32 | |
| Heilongjiang | ENF-H | --- | SEM | 19 ± 4.2 | Yu et al. (2017) |
| | DBF-H | | | 11 ± 5.1 | |
| | DNF-H | | | 20 ± 2.7 | |
| | MBF-H | | | 11 ± 2.0 | |
| | MNF-H | | | 39 ± 7.4 | |
| | MF-H | | | 16 ± 1.9 | |
| Zhejiang | NF-Z | --- | SEM | 23 | Zheng et al. (2019) |
| | BF-Z | | | 15 | |

## 5 Discussion

In this study, we derived ten forest NPP-age curves for six regions and five forest cover types in China based on 3121 forest field samples (Fang et al., 2018) and tested five mathematical models including the SEM function, SDP function, L function, M function, and Γ function for simulating the curves. The SEM function and Γ function performed prominently in fitting all ten curves, nearly perfectly capturing NPP variations with forest age; while for the SDP function, L function, and M function, the NPP-age variations were not well captured. The declining trend of forest NPP in old ages was not captured by the L function and M function, while the SDP function exhibited a sharp decline of NPP to 0 before reaching 200 years in five forest NPP-age curves. These results were consistent with the study that compared NPP-age relationships in boreal and temporal forests constructed using the SDP function, L function, M function, and Γ function (Tang et al., 2014). Further analysis using the normalized NPP-age curves with forest age extended to 300 years suggested that the Γ function tends to force NPP to be zero at old ages for some forest NPP-age curves due to the limited old-aged forest field survey samples. Considering the overall performance with currently available field survey samples, the SEM function was regarded as optimal for building forest NPP-age curves in China.

### 5.1 The mechanism of NPP-age variations

Forest NPP exhibits a rapid increase during young ages, reaching a peak in a middle age, and subsequently declining in old age (Chen et al., 2003; Yu et al., 2017; He et al., 2012). The increase of forest NPP at young ages is mainly driven by a fast increase in leaf area (Ryan et al., 1997; Yu et al., 2014), when the forest stand is relatively open with low competition for light, water, and nutrients (Gower et al., 1996; Yan et al., 2006). Previous studies attributed the decline in NPP in aging forests primarily to the reduction in gross primary productivity (GPP) as the forest ages, while autotrophic respiration (Ra) increases with age (Tatuo KIRA and SHIDEI, 1967; Odum, 1969). However, recent studies have challenged this classical view, revealing that the age-driven decline in NPP is primarily driven by the decrease in both GPP and Ra as forests age, with GPP declining at a faster rate than Ra (Drake et al., 2011; Ryan et al., 1997, 2004; Ryan and Waring, 1992; Tang et al., 2014). This decline in forest NPP during old ages can be attributed to nutrient limitation and ecosystem succession (Camenzind et al., 2018; Fisher et al., 2012; Gao et al., 2018; Gough et al., 2008). As forests age increases, soil nutrients are often depleted to some extent. Trees respond by intensified competition for these nutrients through growing more fine roots to absorb them (Ryan et al., 1997; Tang et al., 2011). This increased competition can lead to nutrient deficiency and decreased NPP. However, old forests can maintain stable growth conditions through self-renewal and continue to accumulate carbon with a magnitude of carbon sinks ranging from 1.0 to 3.2 Mg C ha–1 yr–1 (Gundersen et al., 2021; Luyssaert et al., 2008).

Generally, forest NPP in southern China tends to reach its peak earlier than that in northern China (Yu et al., 2017; Wang et al., 2018; Zheng et al., 2019). This pattern can be attributed to China's wide latitudinal range, resulting in significant variations in temperature and precipitation. Higher temperatures and precipitation contribute to an earlier peak of forest NPP in southern China (Litton et al., 2007; Sillett et al., 2010). Moreover, microscale factors, such as increased hydraulic resistance

for tall trees, diminished nutrient supply, and the contraction of leaf area due to crown abrasion, may also contribute to the decline of NPP at younger ages (Ryan et al., 1997).

## 5.2 Limitations and future modifications

There were also some limitations in this study. First, considering the sample numbers, distributions, and age groups, only ten forest NPP-age curves were derived across the entire China. Except for DNF, the differences in forest NPP-age curves between the southern and northern regions of China (Dai et al., 2011) were considered for all forest cover types. For EBF, its samples were sufficient to separate two forest NPP-age curves in southern China: one is for region SW, and the other is for the regions of S/E. The constructed forest NPP-age curve may not be universally applicable to all areas within the region or specific forest types. For future modifications, it is advisable to incorporate additional samples and develop separate NPP-age curves tailored to smaller regions.

Second, mortality is ignored in this study due to a lack of observations at the ground plots and its average small proportion of NPP except for extreme conditions. According to He et al. (2012), mortality considered in NPP calculations typically includes stand dead trees and down dead wood, which, in the United States, accounted for an average of 3.7% of NPP across 18 forest type groups (Shang et al., 2023). Similarly, in China, mortality varies among different tree species and regions. For example, in Northeast China, 17 major tree species experience a drought-induced mortality rate of 0.49% (Ma et al., 2023). In Fujian Province, according to the 8-th and 9-th NFI data, the average loss rate of forest stock volume due to mortality was 2.5% and 3.49% of the total stock volume, respectively. When converting stock volume into NPP, these proportions attributed from mortality can be even smaller (Zhang et al., 2019), because the calculation of NPP includes additional components such as foliage turnover and fine root turnover in the soil. Nonetheless, in specific environments such as drought, fires, and pest infestations (Shang et al., 2022), the mortality rate of certain tree species can significantly increase, sometimes comprising a substantial proportion of the aboveground NPP (Xu et al., 2012; Ding et al., 2023). Despite this, when constructing the NPP-age curve, these extreme mortality rates were not taken into account, as we mainly focused on the average state across a larger region. Considering the small contribution of mortality to overall NPP and the paucity of ground plot data, mortality was overlooked in this study. Future research efforts could focus on collecting mortality data to enhance the building of the NPP-age curves, and consider the use of NPP-age curves under extreme conditions to simulate variations in forest carbon sequestration during extreme events.

Third, the turnovers of leaves and fine roots, which were also two important components of the field NPP, were calculated based on the assumption that fine root production is linearly correlated with the production of leaves (Litton et al., 2007; He et al., 2012). This assumption was supported by the correlation between new fine root carbon and new leaf carbon indicated by the field measurements (Børja et al., 2008; Burkes et al., 2003; Claus and George, 2005; DesRochers and Lieffers, 2001). It should be noted that fine root production could also be affected by other factors such as soil texture, moisture, and climate (Zerihun and Montagu, 2004), which might be calculated from other carbon allocation methods in future modifications (White et al., 2000).

Fourth, the old-aged forest field survey samples were limited for some forest cover types and regions, resulting in a sharp decrease of forest NPP at old ages for some forest NPP-age curves built by the Γ function. This phenomenon does not deny that Γ function can simulate the relationship between NPP and forest age well in the range of forest age with field survey samples. With more old-aged forest field survey samples collected, the Γ function could also be a good choice for building the forest NPP-age curves and serve as the model inputs to facilitate forest carbon cycle modelling with a process-based model.

Last, the site condition was not considered in building the forest NPP-age curves. It has been shown that the site condition can impact the forest NPP-age variations, and better site conditions can result in faster growth of NPP in young age, greater peak NPP, and steeper decline of NPP in old ages (Yu et al., 2017; Wang et al., 2018). However, the lack of site condition data impeded our ability to build separate forest NPP-age curves according to the site conditions. Regardless of these limitations, this study still provides valuable insights into forest NPP-age variations, and collecting more comprehensive data in the future can further enhance the construction of forest NPP-age curves.

## 6 Conclusions

In this study, we investigated the relationship between forest NPP and age in China by using 3121 forest field survey samples and remote sensing data. Ten forest NPP-age curves were derived for all China's forests based on the spatial distributions of forest cover type, biomass, and age of the field survey data. Five models, including the SEM function, SDP function, L function, M function, and Γ function, were compared to determine the optimal model for building the forest NPP-age curves in China. The comparison against the survey data showed that the SEM function and the Γ function performed much better than the other three models, and through extending forest ages to 300 years, we found that the SEM function was more applicable than the Γ function in capturing the forest NPP-age variations at old ages. Considering the overall performance with currently available field survey samples, the SEM function was regarded as optimal for building forest NPP-age curves in China. The built forest NPP-age curves offer an independent and comprehensive source of information for forest growth estimation and can facilitate forest carbon cycle modelling and future projection in China and elsewhere.

## Code availability

The codes for building forest NPP-age relationships are available by request from the corresponding authors.

## Data availability

The coefficients of the built forest NPP-age relationships are available in Table 2.

## Author contribution

Conceptualization, R.S. and J.M.C.; Methodology, P.L., R.S. and J.M.C.; Validation, P.L.; Formal analysis, P.L., M.X. and X.L.; Writing—original draft, P.L. and R.S.; Writing—review & editing, R.S., J.M.C. and M.X.; Funding acquisition, R.S. and M.X. Data curation, G.Y., N.H. and L.X..

## Competing interests

The authors declare that they have no conflict of interest.

## Acknowledgments

This research was funded by the National Natural Science Foundation of China (42101367 and 42201360), the Natural Science Foundation of Fujian Province (2021J05041), the Fujian Forestry Science and Technology Key Project (2022FKJ03), and the Open Fund Project of the Academy of Carbon Neutrality of Fujian Normal University (TZH2022-02).

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
