# Peer review of "Evaluation of five models for constructing forest NPP-age relationships in China based on 3121 field survey samples"

_Biogeosciences, 2023_

## Author Comment (AC1)

**Comments 1:**

Reliable NPP-age relationships are critical for carbon flux simulations and forest management. In this study, 10 NPP-age curves for different regions and forest types in China were derived based on field and satellite data. The authors also compared the performance of five different math models in deriving these relationships. The results were clearly described. The authors also compared their results to existing models in the same region and differences were interpreted. In my opinion, this study is unique in making use of the large amount of field data and satellite LAI time-series, and credible and up-to-date results provided. I found no major problems in this study. A list of minor comments is provided below for the authors' reference.

**Response:**

Thanks for your positive feedback.

**Comments 2:**

It is my interest to see a discussion on how the $CO_2$ fertilization could have affected in the collected datasets (i. e., biomass inventory and LAI), and correspondingly, how the trend changes in these datasets could be propagated into these derived curves. This is important because of the potentially uneven fertilization effects in different periods of time-series, compared to a scenario in the pre-industrial era. However, this might already be out of the scope of this study since a focus of this study is to compare different math models.

**Response:**

Thanks for your valuable comments and suggestions. We appreciate your insights on $CO_2$ fertilization's effect on biomass and LAI. The primary aim of this study was to establish forest growth curves useful for the entire landmass of China and to compare different models that would be applicable to China as well as other regions. However, the ground survey data we not designed in a way that is conducive to the study of the $CO_2$ fertilization effect (pairs of old and young stands at the same environmental conditions are needed to isolate the effect). We therefore could not achieve this objective in this study, although the $CO_2$ fertilization effect is embedded in the growth curves.

**Comments 3:**

L27: is not it GPP the largest flux (component)?

**Response:**

Thanks for your valuable comments. GPP is undoubtedly the primary component of ecosystem flux, but NPP represents the net production after deducting plant respiration from GPP. To avoid confusion, it was revised.

*"Forest net primary productivity (NPP), which represents the net carbon gain from the atmosphere (Fang et al., 2001; Chapin et al., 2006), constitutes a key component of the terrestrial carbon cycle (Alexandrov et al., 1999; Hasenauer et al., 2004; Zha et al., 2013; Zhao and Zhou, 2005)."*

**Comments 4:**

L29: a gradual decline is not always seen, especially for some mixed forests.

**Response:**

Thanks for your valuable comments. It was revised as "generally featured". Generally, forest NPP tends to decline in old ages. But there are also exceptions in mixed forests, due to their ecological diversity and complexity, the NPP of individual components changes at different rates with forest age, preventing a significant gradual decline in forest NPP in older ages.

*"It varies significantly with forest age (Bond-Lamberty et al., 2004; Wang et al., 2007, 2011), generally featured by an initial increase at young ages, a maximum at a middle age, and then a gradual decline at old ages (Yu et al., 2017; He et al., 2012)."*

**Comments 5:**

Table 1: turnover rate for EBF (evergreen) is "one", is it true?

**Response:**

Thanks for your valuable comments. Due to the lack of data in White et al., (2000), the turnover rate for EBF was previously set equal to that of DBF. After an extensive literature review, we found that the turnover rate for EBF is 0.86 (Lu et al., 2016).

As shown in Response Figure 1, we compared the built NPP-age curves of EBF and MF using the turnover rate of 1 and 0.86 for EBF. It shows that the peak NPP ages from the NPP-age curves remain unchanged for three curves, and only had a change of one year for two curves for EBF (SW, 43 vs. 42) and MF (SW/S/E, 38 vs. 39). In terms of the percentage of NPP duction at the age of 200 years, compared to the original curve, EBF(NE/N/NW) increased by 1.64%, EBF(SW) decreased by 1.63%, EBF(S/E) increased by 0.85%, MF(NE/N/NW) increased by 0.69%, and MF(SW/S/E) decreased by 0.69%.

The related figures (Figures 3-6), tables (Tables 1-2), and descriptions were revised. Despite

these small changes, the main conclusion of this study does not change, and the SEM function was still the optimal model for building NPP-age curves in China.

[Figure]

**Response Figure 1.** Comparison of NPP-age curves fitted with different leaf turnover rates for EBF and MF. The blue colours indicate using a leaf turnover rate of 1, while the red colours indicate using a leaf turnover rate of 0.86. E: Northeast China; N: North China; NW: Northwest China; E: East China; S: South China; SW: Southwest China; EBF: Evergreen Broad-leaved Forests; MF: Mixed Forests.

**Comments 6:**

L99: LAI data in 1981-2022 were used – did you use the LAI in a specific year to calculate the corresponding $L_l$ (age) in the NPP-age curve? LAI(age) may not be available for the earlier stage of old forest (42 yr+), did you use spatial surrogate? This could be clear in the revision.

**Response:**

Thanks for your valuable comments and suggestions. The annual maximum LAI for the survey year of the field sample was used to calculate the corresponding $L_l$ (age) in the NPP-age curve. The field surveys were conducted from 2009 to 2013, and we only used the GLOBMAP Version 3 LAI product within the time period from 2009 to 2013. As we didn't build the relationships between LAI and forest age, the spatial surrogate was not used for old forests (42 yr+). The related descriptions in section 2.2 were revised to be clearer.

*"This dataset records the site location, survey time (from 2009 to 2013), forest cover type, stand age, forest aboveground biomass, forest underground biomass, and so on. These attributes were first used to calculate the forest field NPP and then build the forest NPP-age curves."*

*"According to the site location and survey time, the annual maximum LAI within the survey year for the field survey sample was used to calculate the turnovers of foliage and turnovers of fine roots in the soil."*

**Comments 7:**

L191: "consistent" - not sure for mixed forests

**Response:**

Thanks for your valuable comments. It was revised as "generally consistent". Generally, forest NPP tends to decline in old ages. But there are also exceptions in mixed forests, due to their ecological diversity and complexity, the NPP of individual components changes at different rates with forest age, preventing a significant gradual decline in forest NPP in older ages.

*"Across various forest types, the annual increment of total living biomass rises in early forest development, peaks mid-term, and later declines, generally consistent with the trajectory of NPP with age."*

**Comments 8:**

L208: "the rectangle in each line…" – no rectangles are seen.

**Response:**

Thanks for your valuable comments. The figures were updated from rectangle to triangle, but the revision was missed in the main text, and it was revised now.

*"Figure 6. The normalized NPP-age curves built from the SEM function and the Γ function with the forest age extended to 300 years. The solid lines are for the age period with field data (the triangle in each line indicates the largest age with the field data), and the dashed lines are for the age period without field data. E: Northeast China; N: North China; NW: Northwest China; E: East China; S: South China; SW: Southwest China; EBF: Evergreen Broad-leaved Forests; ENF: Evergreen Needle-leaved Forests; DBF: Deciduous Broad-leaved Forests; DNF: Deciduous Needle-leaved Forests; MF: Mixed Forests."*

**Comments 9:**

Figure 4, panel 8 (DNF …) – for Delta_Bc (biomass increment): any interpretation for the "increase" pattern in 120-180 yrs? Slightly increasing pattern is also observed for panel 1 (EBF).

**Response:**

Thanks for your valuable comments. For DNF, the increase of Delta_Bc in 120-180 years may result from the blending of two different NPP-age curves. As shown in Response Figure 1, the DNF samples aged >100 years were all located in the northwest of China. But due to the limited samples

across different age groups for DNF, we didn't build separate NPP-age curves in the northwest of China but used a single curve nationwide. For EBF (NE/N/NW), the slightly increasing trend of forest NPP in old ages is more likely influenced by noises, as the standard deviation of forest NPP in 100-140 years was much larger than the standard deviation of forest NPP in 40-90 years.

We revised and added the related descriptions, and recommended collecting more data in the future to discern NPP-age curves for different regions more precisely.

*"4.1 Characterization of forest NPP-age curves*

*Figure 3 shows the comparison of the five models in building the NPP-age curves for various forest types and regions in China based on the averaged forest field NPP, and the three components of forest field NPP for each curve are shown in Fig. 4. The annual increment of total living biomass constitutes the predominant share of NPP, markedly surpassing the sum of other components in NPP. Despite their relatively minor proportions, the turnover rates of foliage and the fine roots in the soil are essential components of NPP (He et al., 2012). Across various forest types, the annual increment of total living biomass rises in early forest development, peaks mid-term, and later declines, generally consistent with the trajectory of NPP with age. There are also exceptions for some curves with slightly increasing trends of some NPP components in old ages. This might be explained by the following reasons: first, this study did not separate the overstory and understory LAI, and the presence and growth of understory LAI can influence the trends of the NPP components at old ages; second, due to the limited forest field survey samples, we merged samples over large regions to build the forest growth curves for some forest cover types in China, and this can also be a reason for not showing a declining trend. Lastly, in mixed forests, the growth of different forests is asynchronous, leading to the absence of a declining trend in old ages."*

[Figure]

**Response Figure 2.** Distribution of deciduous needle-leaved forests (DNF) field survey sites and their age groups (different colour indicates different age groups).

**Comments 10:**

Figure 4 again: increase, stable, and decrease patterns of LAI (therefore for leave and fine-root biomass) are seen. It would be interesting to see the interpretation of these patterns (trends).

**Response:**

Thanks for your valuable comments. Indeed, Figure 4 illustrates both a declining trend in the forest LAI with age, as well as stable and increasing trends of forest LAI with age. This might be explained by the following three reasons: Firstly, the LAI used in this study did not differentiate between overstory and understory LAI. The presence and growth of understory LAI can influence the broader trend of the forest LAI's decline as the forest ages. Secondly, due to the limited forest field survey samples, we merged samples over large regions to build the forest growth curves for some forest cover types in China, and this can also be a reason why the LAI in older forests did not show a declining trend. Lastly, in mixed forests, the growth of different forests is asynchronous, which could also contribute to the absence of a declining trend in LAI among old forests. The above descriptions were also added to the manuscript.

*"Across various forest types, the annual increment of total living biomass rises in early forest development, peaks mid-term, and later declines, generally consistent with the trajectory of NPP with age. There are also exceptions for some curves with slightly increasing trends of some NPP components in old ages. This might be explained by the following reasons: first, this study did not separate the overstory and understory LAI, and the presence and growth of understory LAI can influence the trends of the NPP components at old ages; second, due to the limited forest field survey samples, we merged samples over large regions to build the forest growth curves for some forest cover types in China, and this can also be a reason for not showing a declining trend. Lastly, in mixed forests, the growth of different forests is asynchronous, leading to the absence of a declining trend in old ages."*

**Comments 11:**

Figure 5: this figure can be replaced by a Table, with max/min numbers bolded, but this is up to the authors. Unit for RMSE needs to be added.

**Response:**

Thanks for your valuable suggestions, but we think that figures allow readers to see the relative performance of different models at a glance. The unit for RMSE ($gC\ m^{-1}\ yr^{-1}$) was added in the

revised Figure 5.

**Comments 12:**

Table 2: either coefficient "a" has a unit, or the unit of derived total NPP needs to be indicated.

**Response:**

Thanks for your valuable suggestions. The units were added in the revised Table 2.

**Comments 13:**

Table 3: for the "Source – This study", are these peak-ages derived from NPP-age curves, or from measured data (Fig. 3)? It will be useful to show/interpret any differences (e.g. 39 yrs vs 50 ys for panel 10 – MF in Fig. 3?).

**Response:**

Thanks for your valuable comments. They were derived from the built NPP-age curves, and "*This study*" has been changed to "*Our NPP-age curves*".

To smooth out short-term fluctuations and better represent the overall trend over a period, we averaged the NPP values for specific age ranges. However, compared to the original dataset, this approach might shift the age at which the highest NPP is observed in the averaged dataset. If the peak is pronounced or lies close to the boundaries of the averaging interval, this process can broaden the peak. The discrepancy between 39 years and 50 years for panel 10 in Figure 3 arises because averaging the highest NPP values in the original dataset, which might be concentrated in a specific age range, while surrounding values can alter the age of the peak in the fitted curve. Upon analyzing the original NPP, we noted that higher NPP values were evenly distributed between ages 30-50. The fitted NPP curve, however, pinpointed a peak forest age of 39 years, which aligns more closely with the original NPPs.

---

## Author Comment (AC2)

**Response to reviewer #2's comments**

**Comments 1:**

This ms. tested different methodologies to fit age-NPP curves over Chinese forests. I think the topic is important and is worth to investigate. Overall, this ms. provides a clear route from reasoning, model testing and analysis, and corresponding comparison with previous studies and explanations of their results. However, I feel substantial improvements are needed before it can be accepted. First of all, the underlying mechanism of age-NPP relationship should be clearly explained and the results should be further explored and explained. It is important to present to the readers the theoretical architechture of the age-NPP relationships and the potential factors that affect these relationships at the very beginning of the ms.. Then the interpretation of the results can be very well linked to the theories or possible mechanisms.

**Response:**

Thanks for your positive feedback. The mechanism of the age-NPP relationship is now added to the introduction section.

*"1 Introduction*

*Forests play a critical role in sequestering atmospheric carbon dioxide (Hicke et al., 2007; Liu et al., 2012; Eggleston et al., 2006; Pan et al., 2011) and mitigating climate change (Friedlingstein, 2020). Forest net primary productivity (NPP), which represents the net carbon gain from the atmosphere in the form of biomass accumulation (Fang et al., 2001; Chapin et al., 2006), constitutes a key component of the terrestrial carbon cycle (Alexandrov et al., 1999; Hasenauer et al., 2004; Zha et al., 2013; Zhao and Zhou, 2005). It varies significantly with forest age (Bond-Lamberty et al., 2004; Wang et al., 2007, 2011), generally featured by an initial increase at young ages, a maximum at a middle age, and then a gradual decline at old ages (Yu et al., 2017; He et al., 2012). The increase of forest NPP at young ages is mainly driven by fast increase of leaf area (Ryan et al., 1997; Yu et al., 2014), while the decline of NPP at old ages is primarily driven by the decrease in both gross primary productivity (GPP) and autotrophic respiration (Ra) as forests age, with GPP declining faster than Ra (Drake et al., 2011; Ryan et al., 1997, 2004; Ryan and Waring, 1992; Tang et al., 2014). These forest NPP-age variations have been integrated into process-based models such as InTEC (Integrated Terrestrial Ecosystem Carbon model) (Chen et al., 2000; Chen et al., 2003; Wang et al., 2011; Zhang et al., 2012) for modelling the forest carbon cycle, and building forest NPP-age curves as model inputs is therefore essential to facilitate forest carbon cycle modelling (Luyssaert et al., 2008; Chen et al., 2000; Zhang et al., 2012; Shang et al., 2023)."*

**Comments 2:**

Secondly, I'm concerning about the curve building as the data is highly accumulated at the young age end so it is questionable that whether the curves are still effective for those older forests, especially when the authors used the extension of the models to old forests to select the best model. This question has existed for long in the discussion of age effect on forest C sequestration, and with the existing dataset, it is hard to say which pattern is good or not. I would suggest the authors to cut the curves to an age range that can be reasonable covered by the data. Alternatively, one needs to provide additional evidence to indicate the decline or flat pattern of NPP along the age gradient is true for old forest. Finally, there are places of presentations and organizations that require essential refinements (see specific comments).

**Response:**

Thanks for your valuable comments and suggestions. Indeed, due to the limited old-aged forest survey samples, the model's comparisons may not be objective for some forest NPP-age curves. Therefore, we have revised the corresponding descriptions and included a paragraph to discuss the related issues in the discussion section.

[revised manuscript text omitted]

**Comments 3:**

Line 20-21: I'm not convinced about the statement on the point of as little samples from the old forests were used in this study and it is highly unknown that to what point NPP stops declining, or whether they can maintain constant over an age of hundreds of years.

**Response:**

Thanks for your valuable comments and suggestions. This statement was revised.

*"**Abstract.** Forest net primary productivity (NPP), representing the net carbon gain from the atmosphere, varies significantly with forest age. Reliable forest NPP-age relationships are essential for forest carbon cycle modelling and prediction. These relationships can be derived from forest inventory or field survey data, but it is unclear which model is the most effective for simulating forest NPP variation with age. Here, we aim to establish NPP-age relationships for China's forests based on 3121 field survey samples. Five models, including the Semi-Empirical Mathematical (SEM) function, the Second-Degree Polynomial (SDP) function, the Logarithmic (L) function, the Michaelis-Menten (M) function, and the $\Gamma$ function, were compared against field data. Results of the comparison showed that the SEM and the $\Gamma$ function performed much better than the other three models.* But due to the limited field survey samples at old ages, the $\Gamma$ function showed a sharp decrease in NPP (decreased to almost zero) at old ages when building some forest NPP-age curves, while SEM could reasonably capture the variations of *forest NPP at old ages. Considering the overall performance with currently available forest field survey samples, SEM was regarded as the optimal NPP-age model. The finalized forest NPP-age curves for five forest types in six regions of China can facilitate forest carbon cycle modelling and future projection by using the process-based model (InTEC) in China and may also be useful for other regions."*

**Comments 4:**

Line 22: The final statement is vague to me. What is "forest carbon modelling" and "future carbon projections".

**Response:**

Thanks for your valuable comments. It was revised.

The NPP-age relationships are important inputs of the process-based model (InTEC) (Chen et al., 2000; Zhang et al., 2012; Shang et al., 2023) for forest carbon cycle modelling and future projection. We have changed the phrase to "forest carbon cycle modeling and future projection". "forest carbon cycle modelling" refers to modelling the full carbon cycle for forest ecosystems including photosynthetic carbon uptake, autotrophic and heterotrophic respiration, regrowth after disturbance and direct carbon release during a disturbance. "forest carbon cycle future projection" refers to predicting these carbon cycle components and carbon sinks according to future climate

scenarios.

*"**Abstract.** Forest net primary productivity (NPP), representing the net carbon gain from the atmosphere, varies significantly with forest age. Reliable forest NPP-age relationships are essential for forest carbon cycle modelling and prediction. These relationships can be derived from forest inventory or field survey data, but it is unclear which model is the most effective for simulating forest NPP variation with age. Here, we aim to establish NPP-age relationships for China's forests based on 3121 field survey samples. Five models, including the Semi-Empirical Mathematical (SEM) function, the Second-Degree Polynomial (SDP) function, the Logarithmic (L) function, the Michaelis-Menten (M) function, and the $\Gamma$ function, were compared against field data. Results of the comparison showed that the SEM and the $\Gamma$ function performed much better than the other three models. But due to the limited field survey samples at old ages, the $\Gamma$ function showed a sharp decrease in NPP (decreased to almost zero) at old ages when building some forest NPP-age curves, while SEM could reasonably capture the variations of forest NPP at old ages. Considering the overall performance with currently available forest field survey samples, SEM was regarded as the optimal NPP-age model. The finalized forest NPP-age curves for five forest types in six regions of China can facilitate forest carbon cycle modelling and future projection by using the process-based model (InTEC) in China and may also be useful for other regions."*

Chen, W., Chen, J., Cihlar, J.: An integrated terrestrial ecosystem carbon-budget model based on changes in disturbance, climate, and atmospheric chemistry. Ecol. Modell. 135, 55–79. https://doi.org/10.1016/S0304-3800(00)00371-9, 2000.

Shang, R., Chen, J.M., Xu, M., Lin, X., Li, P., Yu, G., He, N., Xu, L., Gong, P., Liu, L., Liu, H., Jiao, W.: China's current forest age structure will lead to weakened carbon sinks in the near future. Innov. 100515. https://doi.org/10.1016/j.xinn.2023.100515, 2023.

Zhang, F., Chen, J.M., Pan, Y., Birdsey, R.A., Shen, S., Ju, W., He, L.: Attributing carbon changes in conterminous U.S. forests to disturbance and non-disturbance factors from 1901 to 2010. J. Geophys. Res. Biogeosciences 117, 1–18. https://doi.org/10.1029/2011JG001930, 2012.

**Comments 5:**

Line 25-27: The statements here are vague again – how you define the "biomass carbon gain from atmosphere" and "the largest component of the terrestrial carbon cycle". It seems applicable for some other terms also but is not totally right as well.

**Response:**

Thanks for your valuable comments. They were revised.

*"Forest net primary productivity (NPP), which represents the net carbon gain from the*

*atmosphere in the form of biomass accumulation (Fang et al., 2001; Chapin et al., 2006), constitutes a key component of the terrestrial carbon cycle (Alexandrov et al., 1999; Hasenauer et al., 2004; Zha et al., 2013; Zhao and Zhou, 2005).”*

**Comments 6:**

Line 30: Again what is "forest carbon modelling"? I would suggest to go through the ms. and revise all those vague terms used.

**Response:**

Thanks for your valuable comments and suggestions. The relevant descriptions were revised.

*"These forest NPP-age variations have been integrated into process-based models such as InTEC (Integrated Terrestrial Ecosystem Carbon model) (Chen et al., 2000; Chen et al., 2003; Wang et al., 2011; Zhang et al., 2012) for modelling the forest carbon cycle, and building forest NPP-age curves as model inputs is therefore essential to facilitate forest carbon cycle modelling (Luyssaert et al., 2008; Chen et al., 2000; Zhang et al., 2012; Shang et al., 2023).”*

**Comments 7:**

Line 31: Unique -> varied.

**Response:**

Thanks for your valuable comments and suggestions. They were revised.

*"Forest NPP-age curves differ considerably for different regions and forest types due to their varied compositions and diverse growth environments (Yu et al., 2017; He et al., 2012).”*

**Comments 8:**

Line 31-46: I would introduce the mechanisms driving the relationship, rather than the data gaps here. Simply pointing out the data gaps in China to me is not a strong motivation of this study. I mean, understanding the patterns is much more important and interesting than fitting models and say which one is the best, in the context of science.

**Response:**

Thanks for your valuable comments and suggestions. Many studies have already investigated the variations of forest NPP with age, and have elucidated the mechanisms behind these variations (Yu et al., 2017; He et al., 2012; Wang et al., 2011). But there is still debate regarding whether old-growth forests act as carbon-neutral entities or carbon sinks, and the extent of their carbon sequestration capabilities (Ryan et al., 1997; Yu et al., 2014). Due to the limited forest field samples at old ages, it is challenging to comprehensively address these debates in this study. Besides, the

aim of this study is to compare and find an optimal model for building the NPP-age curves in China, which will serve as an input for the InTEC model and facilitate China's forest carbon modelling. In our future work, we will collect more old-aged forest field survey samples and try to understand these patterns.

According to your suggestion, we have modified the paragraph as follows:

*"Forest NPP-age curves differ considerably for different regions and forest types due to their varied compositions and diverse growth environments (Yu et al., 2017; He et al., 2012). In Europe (Zaehle et al., 2006), Canada (Chen et al., 2003), and America (Guo et al., 1955; He et al., 2012), forest NPP-age curves have been established for different forest types or regions. However, these curves can't be directly used for China's forest carbon modelling because of regional differences in environmental conditions. The NPP-age curves produced in previous studies have very different and sometimes inconsistent shapes, making it difficult to analyze the influence of environmental conditions on the curves of different forest types. To address these issues, some studies have tried to build forest NPP-age curves in China. Yu et al. (2017) established forest NPP-age curves for twelve major forest types in Heilongjiang province using forest inventory data and yield tables. Wang et al. (2018) derived forest NPP-age curves for nine pure forest types with different site indices within Heilongjiang province using the yield tables, biomass equations, and forest inventory data. Zheng et al. (2019) built two forest NPP-age curves separately for coniferous and broad-leaved forests in Zhejiang province using forest inventory data. But these curves are limited to the provincial level (currently only available in Heilongjiang and Zhejiang provinces), and cannot represent the diverse growth status of China's forests. Wang et al. (2011) constructed five forest NPP-age curves for five representative forest ecosystems in China, but the NPP data used to build these curves were obtained from the simulations of the BEPS (Boreal Ecosystem Productivity Simulator) model (Chen et al., 2012; Ju et al., 2006; Liu et al., 2002, 1999), not forest inventory data or field survey data. Furthermore, these curves didn't consider the significant differences in forest and climate conditions between the north and south of China and were insufficient to differentiate the north-south variations in China (Dai et al., 2011). Therefore, it is essential to develop forest NPP-age curves for the entire China with consideration of the differences in regions and forest types."*

**Comments 9:**

Line 64-67: reads like methodology rather than an introduction of the aims. Also I feel the aim of this study is to understand the underlying patterns, rather than comparing things..

**Response:**

Thanks for your valuable comments. It was revised.

*"There are two objectives of this study: (1) to build forest NPP-age relationships for the entire China considering differences in regions and forest types based on forest field survey data and remote sensing data, and (2) to compare five models and determine the optimal model in building forest NPP-age relationships across China. The built forest NPP-age curves from the optimal model for different forest types and regions in China would be served as inputs of a process-based model to facilitate China's forest carbon cycle modelling and future projection."*

**Comments 10:**

Line 70: rephrase the sentence to something like: Chinese forests consist of five main functional types...

**Response:**

Thanks for your valuable comments and suggestions. It was deleted in the introduction section, and the related descriptions were moved to "2.1 Study area".

*"**2.1 Study area***

*China is selected as the study area, and its forests consist of five mean functional types: Evergreen Broad-leaved Forests (EBF), Evergreen Needle-leaved Forests (ENF), Deciduous Broad-leaved Forests (DBF), Deciduous Needle-leaved Forests (DNF), and Mixed Forests (MF). The five forest types were separated in building forest NPP-age curves with consideration of their different physiological and ecological characteristics (Wang et al., 2011). Except for forest cover types, climatic differences in different regions of China can also affect the forest NPP-age relationships (Li and Zhou, 2015; Song et al., 2018), so regions also need to be divided when building the forest NPP-age curves. According to China's geographical division (Fang et al., 2001), the study area was divided into six regions (Fig. 1): Northeast China (NE), North China (N), Northwest China (NW), East China (E), Southwest China (SW), and South China (S). Significant differences in forest types can be observed among different regions. Region NE (including Heilongjiang, Jilin, and Liaoning provinces) is a typical boreal forest in the world and the most significant natural forest area in China. Region N (including Beijing and Tianjin cities, and Hebei, Shanxi, and Inner Mongolia provinces) accounts for 14% of China's total forest area and is mainly composed of DBF and ENF. Regions NW (including Gansu, Ningxia, Qinghai, Shanxi, and Xinjiang provinces) only account for 2.57% of the total forest area in China. Region E (including Shanghai City and Jiangsu, Zhejiang, Anhui, Fujian, Jiangxi, Shandong, and Taiwan provinces) accounts for 14% of China's total forest area, and its forests show significant zonal characteristics. Region SW (including Yunnan, Sichuan, Xizang, Guizhou, and Chongqing provinces) is the second-largest natural forest area in China, accounting for 26% of China's total forest area and 43% of China's forest stock (Liu et al., 2021). Region S (including Henan, Hubei, Hunan, Guangdong, Guangxi, and Hainan provinces) accounts for 20% of the total forest area in China with a large proportion*

*of planted forests."*

**Comments 11:**

Line 72: okay but it's weird to mention two classification system together with little logical linkage.

Line 70 – 84: I feel the entire paragraph should be reorganized in a better form. As I said above, there should be necessary logical linkages between sentences.

**Response:**

Thanks for your valuable comments and suggestions. It was revised.

*"2.1 Study area*

*China is selected as the study area, and its forests consist of five mean functional types: Evergreen Broad-leaved Forests (EBF), Evergreen Needle-leaved Forests (ENF), Deciduous Broad-leaved Forests (DBF), Deciduous Needle-leaved Forests (DNF), and Mixed Forests (MF). The five forest types were separated in building forest NPP-age curves with consideration of their different physiological and ecological characteristics (Wang et al., 2011). Except for forest cover types, climatic differences in different regions of China can also affect the forest NPP-age relationships (Li and Zhou, 2015; Song et al., 2018), so regions also need to be divided when building the forest NPP-age curves. According to China's geographical division (Fang et al., 2001), the study area was divided into six regions (Fig. 1): Northeast China (NE), North China (N), Northwest China (NW), East China (E), Southwest China (SW), and South China (S). Significant differences in forest types can be observed among different regions. Region NE (including Heilongjiang, Jilin, and Liaoning provinces) is a typical boreal forest in the world and the most significant natural forest area in China. Region N (including Beijing and Tianjin cities, and Hebei, Shanxi, and Inner Mongolia provinces) accounts for 14% of China's total forest area and is mainly composed of DBF and ENF. Regions NW (including Gansu, Ningxia, Qinghai, Shanxi, and Xinjiang provinces) only account for 2.57% of the total forest area in China. Region E (including Shanghai City and Jiangsu, Zhejiang, Anhui, Fujian, Jiangxi, Shandong, and Taiwan provinces) accounts for 14% of China's total forest area, and its forests show significant zonal characteristics. Region SW (including Yunnan, Sichuan, Xizang, Guizhou, and Chongqing provinces) is the second-largest natural forest area in China, accounting for 26% of China's total forest area and 43% of China's forest stock (Liu et al., 2021). Region S (including Henan, Hubei, Hunan, Guangdong, Guangxi, and Hainan provinces) accounts for 20% of the total forest area in China with a large proportion of planted forests."*

**Comments 12:**

Line 184-191: Not a result literally – seems a combination of methodology and figure head.

Line 192 – 195: Method again.

**Response:**

Thanks for your valuable suggestions. They were revised.

*"Figure 3 shows the comparison of the five models in building the NPP-age curves for various forest types and regions in China based on the averaged forest field NPP, and the three components of forest field NPP for each curve are shown in Fig. 4."*

*"Fig. 5 shows the quantitative comparison of the five functions in building forest NPP-age curves across varied forest cover types and regions in China."*

**Comments 13:**

Line 207-209: Seems like something in figure head again.

**Response:**

Thanks for your valuable comments. It was deleted.

**Comments 14:**

Line 215: The reasoning here is very tricky to me – why you think "decreased more than 50 % in old ages" is a useful gauge to identify whether it is reasonable or not. I would expect that at least there are some numbers of corresponding percentages that can be extracted from the previous studies cited here (so you can say okay NPP dropping too fast with age might be unrealistic). But again, as I said earlier, the extension of the fittings to an age of hundreds itself is not that effective becuz little evidence is there.

**Response:**

Thanks for your valuable comments. It was revised, and a discussion about the Γ function was added in the section of discussion.

*"For the two curves of EBF(SW) and DBF(SW/S/E) where the Γ function had the highest R2 and lowest RMSE, the forest NPP decreased to almost zero when the stand age reached 300 years. The forest NPP in the curves of ENF(SW/S/E), MF(SW/S/E), DBF(NE/N/NW), and DNF(CHN) built by the Γ function also decreased sharply at the age of 300 years and decreased to almost zero in the ages of 400-500 years. These forest growth patterns contradict the results of previous studies, which indicated that forest NPP is usually reduced to about half (Mund et al., 2002; Ryan et al., 2004) or one-third (Luyssaert et al., 2008; Wang et al., 2011) of its maximum value. The curves*

*from the Γ function suggest that forests would stop growth completely at old ages and act as carbon sources. However, studies have demonstrated that old forests still act as carbon sinks, despite the controversial magnitude of the forest carbon sink ranging from 1.0 to 3.2 Mg C ha$^{-1}$ yr$^{-1}$ (Gundersen et al., 2021; Luyssaert et al., 2008). Ecologically, we would expect old forests to maintain stable conditions through self-renewal processes, such as the generation of new trees after the mortality of old trees (Harmon et al., 1990). The SEM function that produces stable NPP at old ages is therefore more reasonable in capturing the forest NPP-age variations during old ages, and was determined as the optimal forest model for building the forest NPP-age curves in China (the model coefficients of the built ten curves are provided in Table 2)."*

*"**5.2 Limitations and future modifications***

*……*

*   Third, the old-aged forest field survey samples were limited for some forest cover types and regions, resulting in a sharp decrease of forest NPP at old ages for some forest NPP-age curves built by the Γ function. This phenomenon does not deny that Γ function can simulate the relationship between NPP and forest age well in the range of forest age with field survey samples. With more old-aged forest field survey samples collected, the Γ function could also be a good choice for building the forest NPP-age curves and serve as the model inputs to facilitate forest carbon cycle modelling with a process-based model."*

**Comments 15:**

For the entire results section, I suggest the authors to reorgnize it in a way of "telling story", rather than "presenting a stack of stats".

**Response:**

Thanks for your valuable comments. The results section was revised and re-organized.

*"**4 Results***

*4.1 Characterization of forest NPP-age curves*

*……*

*4.2 Comparison of five models in building forest NPP-age curves*

*……*

*4.3 Comparison to the forest NPP-age curves built previously in China*

*……"*

**Comments 16:**

Fig 3 and Fig 4: I guess there are some bin processings but they are not explicitly introduced in the methods.

**Response:**

Thanks for your valuable comments. They were added in the methods section.

*"To reduce the influence of noises or outliers in building forest NPP-age curves, the forest field NPPs were averaged within different age bins (e.g. 3, 5, 10, or 20 years). The age bins were divided according to the number of samples in each age bin, and if there were not enough samples for some ages, larger age bins would be used."*

**Comments 17:**

Line 261: I would strongly encourage the authors to expand their discussion from an asepct of the underlying mechanisms and how such mechanisms may cause different/similar patterns of age-NPP curves between PFTs, or how the mechanisms may determine the success or failure of the models. As I pointed out earlier, this study should become way more interesting than the current form, if it tells me, for example, why some PFTs reach NPP peaks ealier than the others and why NPP of some PFTs declines faster than the others. Considering the well fitness of the selected models against the massive data, it is really worth to do so. This interpretation would then lead to a more meaningful comparison between the curves from this study and the ones previously built (and potentially facilitate the illustration of the advantage and uniqueness of this study).

**Response:**

Thanks for your valuable comments. It was added in the discussion section.

*"5.1 The mechanism of NPP-age variations*

*Forest NPP exhibits a rapid increase during young ages, reaching a peak in a middle age, and subsequently declining in old age (Chen et al., 2003; Yu et al., 2017; He et al., 2012). The increase of forest NPP at young ages is mainly driven by a fast increase in leaf area (Ryan et al., 1997; Yu et al., 2014), when the forest stand is relatively open with low competition for light, water, and nutrients (Gower et al., 1996; Yan et al., 2006). Previous studies attributed the decline in NPP in aging forests primarily to the reduction in gross primary productivity (GPP) as the forest ages, while autotrophic respiration (Ra) increases with age (Tatuo KIRA and SHIDEI, 1967; Odum, 1969). However, recent studies have challenged this classical view, revealing that the age-driven decline in NPP is primarily driven by the decrease in both GPP and Ra as forests age, with GPP declining at a faster rate than Ra (Drake et al., 2011; Ryan et al., 1997, 2004; Ryan and Waring, 1992; Tang et al., 2014). This decline in forest NPP during old ages can be attributed to nutrient limitation and ecosystem succession (Camenzind et al., 2018; Fisher et al., 2012; Gao et al., 2018;*

*Gough et al., 2008). As forests age increases, soil nutrients are often depleted to some extent. Trees respond by intensified competition for these nutrients through growing more fine roots to absorb them (Ryan et al., 1997; Tang et al., 2011). This increased competition can lead to nutrient deficiency and decreased NPP. However, old forests can maintain stable growth conditions through self-renewal and continue to accumulate carbon with a magnitude of carbon sinks ranging from 1.0 to 3.2 Mg C ha$^{-1}$ yr$^{-1}$ (Gundersen et al., 2021; Luyssaert et al., 2008).*

*Generally, forest NPP in southern China tends to reach its peak earlier than that in northern China (Yu et al., 2017; Wang et al., 2018; Zheng et al., 2019). This pattern can be attributed to China's wide latitudinal range, resulting in significant variations in temperature and precipitation. Higher temperatures and precipitation contribute to an earlier peak of forest NPP in southern China (Litton et al., 2007; Sillett et al., 2010). Moreover, microscale factors, such as increased hydraulic resistance for tall trees, diminished nutrient supply, and the contraction of leaf area due to crown abrasion, may also contribute to the decline of NPP at younger ages (Ryan et al., 1997)."*